# Lysozyme Amyloid Fibril Structural Variability Dependence on Initial Protein Folding State

**DOI:** 10.3390/ijms23105421

**Published:** 2022-05-12

**Authors:** Kamile Mikalauskaite, Mantas Ziaunys, Vytautas Smirnovas

**Affiliations:** Institute of Biotechnology, Life Sciences Center, Vilnius University, LT-10257 Vilnius, Lithuania; kamile.mikalauskaite@gmc.vu.lt (K.M.); mantas.ziaunys@gmc.vu.lt (M.Z.)

**Keywords:** amyloid, protein aggregation, lysozyme, fibril

## Abstract

Amyloid fibril formation is associated with several amyloidoses, including neurodegenerative Alzheimer’s or Parkinson’s diseases. The process of such fibrillar structure formation is still not fully understood, with new mechanistic insights appearing on a regular basis. This, in turn, has limited the development of potential anti-amyloid compounds, with only a handful of effective cures or treatment modalities available. One of the multiple amyloid aggregation factors that requires further examination is the ability of proteins to form multiple, structurally distinct aggregates, based on the environmental conditions. In this work, we examine how the initial folding state affects the fibrilization of lysozyme—an amyloidogenic protein, often used in protein aggregation studies. We show that there is a correlation between the initial state of the protein and the aggregate formation lag time, rate of elongation, resulting aggregate structural variability and dye-binding properties, as well as formation lag time and rate of elongation.

## 1. Introduction

Protein aggregation in the form of amyloid fibrils is linked with the onset and progression of several amyloidoses [1], including neurodegenerative Alzheimer’s or Parkinson’s disease [2]. Despite decades of intense research and countless studies [3], a full understanding of such aggregate formation remains elusive [4]. This, in turn, has significantly complicated the search for potential anti-amyloid compounds and there are currently very few drugs or treatment modalities available [5,6]. The constantly increasing number of affected individuals and projected further growth in cases [7,8] make it vitally important to obtain a better understanding of the amyloid fibril formation process.

In general, the mechanism of protein aggregation can be broken down into multiple basic parts. First, proteins have to undergo a misfolding event, during which they form the initial amyloid nuclei (primary nucleation) [9,10]. These nuclei then grow by incorporating other homologous protein molecules into their structure, forming elongated aggregates—fibrils [11,12,13]. Once sufficiently sized aggregates form, secondary processes can also occur during the elongation phase, which include fibril fragmentation [14] and surface-mediated nucleation (fibril surface acts as a catalyst for new nuclei formation) [15,16,17]. Apart from this general mechanism, there are also multiple additional events proposed, such as fibril maturation over a long time period [18], as well as lateral association and aggregate cluster formation [19].

The structure of amyloid fibrils has been studied using a wide range of techniques. These include solid state nuclear magnetic resonance (ssNMR), cryo-electron microscopy (cryo-EM), atomic force microscopy (AFM), Fourier-transform infrared spectroscopy (FTIR), as well as small-angle X-ray scattering (SAXS). These methods have also been utilized to track and analyse the formation of amyloid fibrils and to determine the various types of intermediate species present during the aggregation reaction [20,21,22,23]. While most protein aggregation events follow a similar mechanism, there is a wide variety of structurally and morphologically distinct fibrils formed, depending on the initial protein and the reaction conditions, such as ionic strength, pH, temperature or protein concentration [24,25,26,27,28]. It was even shown that amyloid-beta (which is associated with Alzheimer’s disease) [29], alpha-synuclein (Parkinson’s disease) [18], prion proteins (prionopathies) [30] and Tau proteins (tauopathies) [31] can aggregate into multiple distinct structures under the same environmental conditions. Structural polymorphism is considered to be related to fibril toxicity, replication rates and could explain the onset of different diseases, originating from the same precursor protein [32]. While the ability of identical proteins to form conformationally distinct amyloid fibrils is an interesting phenomenon, it significantly complicates the search for potential anti-amyloid compounds or treatments.

We recently observed that prion protein polymorphism depends on the folding state of the initial protein, with distinct types of fibril structures being formed above and below the melting temperature [30]. Similarly, the formation of aggregates with higher cytotoxic properties at higher temperatures [33] was reported for lysozyme—a model amyloidogenic protein, commonly used in aggregation studies [34]. Considering that temperature is often modulated to alter the rate of protein aggregation in vitro [35,36] and that lysozyme amyloid formation is known to have a condition–structure relationship [37,38], it was important to examine the phenomenon in greater detail. In this work, large sets of identical hen egg-white lysozyme samples were aggregated under a range of temperatures, from lower, where most protein molecules were folded, to higher, where they were fully unfolded. The generated aggregates were then examined and compared based on their secondary structure and dye-binding properties. We show that lysozyme amyloid fibril polymorphism exists both below and above the melting temperature and that there is a significant shift in the variability of structures associated with the folding state of the initial, non-aggregated protein.

## 2. Results

All details regarding sample preparation, experiments and data analysis are reported at the end of the article. In order to determine the melting point of lysozyme under the selected experimental conditions (phosphate-buffered saline (PBS) solution, containing 2 M guanidine hydrochloride, pH 7.4) and select a range of temperatures for the aggregation experiments, an 8-anilinonaphthalene-1-sulfonic acid (ANS) protein melt assay was carried out (Figure 1A). Up to 50 °C, it was observed that the dye signal intensity gradually decreased due to a temperature-related reduction in fluorescence quantum yield. After this point, the ANS emission intensity began rising to a maximum point at 65 °C, after which it rapidly decreased. These results show that the majority of lysozyme was folded up to 50 °C and it is fully unfolded at 65 °C (T_m_ = (57 ± 1) °C). Based on this information, four different temperature conditions were chosen for further examination (marked with colour-coded dashed lines in Figure 1A).

For each temperature, full 96-well plates of identical lysozyme samples were incubated under constant agitation (as described in the Materials and Methods section) and their fibrillization reactions were tracked by measuring the fluorescence emission intensity of fibril-bound thioflavin-T (ThT) (examples of aggregation reaction kinetic curves are shown in Appendix B Figure A1). At 50 °C, the aggregation lag time (t_lag_) had an average value of ~1300 min (Figure 1B), and a temperature increase of 5 °C caused a 3-fold decrease in t_lag_ to ~400 min. Further changes to the reaction temperature had a less significant effect on the t_lag_ value (~190 min at 60 °C and ~160 min at 65 °C), with a point of discontinuity detected between 55 °C and 60 °C, which was similar to the T_m_ value of lysozyme under these experimental conditions. The apparent rate constant of aggregation had the most significant change between the two lowest tested temperatures (2-fold increase from ~0.01 to ~0.02), while the values between 55 °C and 65 °C followed a linear trend in a semi-logarithmic plot (Figure 1C). Similar to the lag time values, there was also a point of discontinuity; however, in this case, it was detected closer to 55 °C (slightly below the T_m_ value). Despite the massive differences in both t_lag_ and apparent rate constant values in this temperature range, there were no detectable sub-groups of samples with specific aggregation kinetic parameters and all data followed a normal distribution.

During the aggregation assay, a substantial difference between the end-point ThT fluorescence emission intensity values of samples generated at low and high-temperature conditions was observed. There was also a significant variation in these values between samples from the same aggregation conditions. Such distinct bound-ThT fluorescence intensities can be associated with the existence of fibrils with different secondary structures or morphologies [30,39]. These ThT-binding modes are usually accompanied by specific maximum excitation and emission wavelength positions, which can be used to differentiate between fibril types [40].

Each sample‘s excitation/emission matrix (EEM) was scanned and their “center of mass“ positions, as well as their correlation to the signal intensity, were compared. In the case of lysozyme aggregates prepared at 50 °C and 55 °C, the EEM maxima positions were clustered at a specific area (449–451 nm excitation and 480–483 nm emission wavelengths), with only a small amount of samples deviating from this position (Figure 2A,B). When the aggregation temperature increased past the T_m_ value, a sizeable portion of sample EEM maxima positions deviated from the main cluster, with certain positions being located as far as 5 nm from the cluster excitation or emission wavelength values (Figure 2C). Matters became even more interesting when the majority of initial lysozyme monomers were in their unfolded state during aggregation. In this case, a new main cluster formed at 444–446 nm excitation and 481–483 nm emission wavelengths, with the large dispersion of positions persisting (Figure 2D), as was the case for the 60 °C samples (Figure 2C).

When examining the correlation between sample EEM positions and fluorescence intensity (Figure 2E), the high-intensity samples appeared near the lower-temperature cluster position (observed in Figure 2A,B), while the lower-intensity samples gathered around the higher-temperature cluster position (observed in Figure 2D). There was also a clear dependence between the distribution of sample fluorescence intensity values and the temperature used in their preparation (Figure 2F). When most lysozyme monomers were folded, the resulting ThT fluorescence intensity had an average value of ~440 a.u. with a relatively small dispersion. At 55 °C, the average value decreased to ~360 a.u. and there was a significant increase in the signal intensity deviation. When the aggregation temperature passed the T_m_ value, there was a substantial reduction in average signal intensity (~110 a.u.), while the high level of dispersion persisted. At the highest tested temperature, where most lysozyme monomers were unfolded, the average fluorescence intensity was even lower (~40 a.u.) and the value dispersion mirrored the 50 °C condition samples.

Interestingly, conditions below the T_m_ value did not result in a single EEM position located at the high-temperature cluster, while 60 °C and 65 °C led to samples with positions at both cluster areas. This suggests that specific aggregate structures or morphologies require the initial protein to be unfolded. The fluorescence intensity distributions displayed that all temperature conditions caused the formation of samples with significant deviation from the average values (low-intensity samples at low temperatures and large-intensity samples at high temperatures). This hints at a possibility of mixtures, composed of different types of aggregates, which have specific ThT-binding parameters.

Since the EEM data suggested a possible high variety of lysozyme aggregate structures, each sample was replicated and examined using Fourier-transform infrared spectroscopy (FTIR). In the case of 50 °C (Figure 3A) and 55 °C (Figure 3B) samples, the FTIR spectra shared similarities, between both different temperature conditions as well as between each other. At 60 °C (Figure 3C), there appeared to be a mixture of distinct secondary structures, with some spectra being similar to those observed at lower temperatures. At the highest temperature (Figure 3D), the variability decreased and most samples had FTIR spectra dissimilar to ones that were present at lower temperatures.

Peak fitting for the FTIR spectra revealed that there were three dominant secondary structures (Figure 3E–J), with distinct peak maxima positions and areas. The Type 1 aggregates were the main structure at 50 °C and 55 °C, while making up only a fraction of 60 °C samples and being nearly non-existent at 65 °C. Conversely, the Type 2 and 3 aggregates were only present at 60 °C and 65 °C temperature conditions. Comparing the peak maxima positions and areas revealed that all three fibril types had similar maximum position peaks at 1614–1615 cm^−1^ (associated with strong hydrogen bonding in the beta-sheet structure [41]) and 1627 cm^−1^ (associated with weaker hydrogen bonding). While the 1614–1615 cm^−1^ position peaks had similar areas among all aggregate types, the 1627 cm^−1^ peaks were significantly smaller in the case of Type 2 and 3 aggregates. Overall, Type 1 had the highest percentage of cross-beta structures, Type 2 had less and Type 3, a minimum amount of such beta sheets. A small peak at 1637 cm^−1^ in the FTIR spectrum of Type 2 aggregates can be attributed to the weak hydrogen bonding of beta sheets, while significant peaks at 1642–1643 cm^−1^ in Type 1 and 3 sample FTIR spectra and at 1648 cm^−1^ in the case of Type 2 aggregates is associated with the presence of unstructured regions. The rest of the peaks in all spectra most probably arise from different turn/loop motifs, though there is a possibility that peaks at the highest wavenumbers could mean the presence of some antiparallel beta sheets.

Since the higher-temperature aggregate FTIR spectra displayed smaller peaks, associated with beta-sheet hydrogen bonding and more unstructured regions, their ability to form fibrillar structures was examined using atomic force microscopy (AFM). In order to facilitate the formation of elongated structures, the samples were reseeded under quiescent conditions at their respective temperatures. For all three aggregate types, the ThT fluorescence intensity signal growth proceeded with no lag phase (Figure 4), indicating aggregation self-replication. In the case of Type 2 and Type 3 aggregates (Figure 4B,C), the end-point fluorescence intensity was significantly lower than Type 1 (Figure 4A), as was observed with the initial samples.

The Type 1 AFM image displayed long (3–5 µm) fibrillar structures, with an average cross-sectional height of 10 nm (Figure 4D). The Type 2 aggregates were significantly shorter (~0.3 µm), with a lower average height of 2 nm and they were associated into web-like systems (Figure 4E). This sample also contained a number of small, round structures, which may be short lysozyme oligomers or amorphous aggregates. Unlike the previous two cases, the Type 3 sample contained mostly short aggregates (Figure 4F), which associated into larger clusters and very few elongated structures were observed. These images suggest that the higher temperature resulted in lysozyme aggregates, which had lower stability and likely formed a considerable number of amorphous structures, coinciding with the significantly reduced beta-sheet-related peaks in their respective FTIR spectra.

In order to determine whether the lower-temperature fibrils were capable of replicating their structure at higher temperatures and vice versa, a reseeding experiment was carried out, as described in the Materials and Methods section. When the Type 1 fibrils were reseeded at 65 °C, the resulting aggregate FTIR spectra was nearly identical to the original (Figure 5A), showing that the fibrils were capable of replicating their structure and that the increased temperature had no effect on them. Conversely, when the Type 2 and 3 aggregates were reseeded at 50 °C, there was a significant change in their FTIR spectra (Figure 5B,C). In both cases, the resulting spectra became similar to the Type 1 aggregates and to one another.

Based on the aggregation kinetic data, the Type 1 fibril reseeding reaction had practically no lag time and proceeded immediately (Figure 5D). This was not the case for both Type 2 and 3 aggregates (Figure 5E,F), where a considerable lag period was observed. Despite the aggregates causing a significant reduction in lag time (~200 min, as opposed to ~1500 min during spontaneous reaction), the change in FTIR spectra suggested that they were not efficient at replicating their structure at 50 °C. This reduced lag time may be due to the Type 2 and 3 aggregates acting as a surface for secondary nucleation, which would explain the formed aggregate FTIR spectra similarity to Type 1 fibrils.

## 3. Discussion

Based on the results from this study, it is clear that the initial folding state of lysozyme is an important factor for multiple amyloid aggregation aspects, including lag time, resulting structural variability, bound-ThT fluorescence intensity and self-replication. In all cases, a discontinuity of kinetic or structural parameters was observed to occur in the relatively small gap between temperatures where lysozyme transitioned from folded to unfolded states. The complex effect of this transition has to be taken into account during studies of lysozyme amyloid aggregation.

The first notable aspect was the significant temperature-related shift in aggregate secondary structure. At lower temperatures, the sample FTIR spectra displayed a considerably larger content of beta-sheet structures, when compared to the two different aggregate types formed above the melting temperature. This transition, however, did not occur instantly, as we observed Type 1 fibrils present at higher-temperature conditions as well, with their number decreasing significantly when lysozyme was unfolded. Interestingly, there was a minor level of structural variability, even under identical conditions, most visible based on the FTIR spectra peak area value deviations (Figure 3H–J). Such variability was also previously observed for prion protein amyloid fibrils [30]. These results suggest that when lysozyme is (partially) folded, it can only transition to one dominant amyloid aggregate (Type 1, Figure 3E), with a minor level of structural variability. However, once lysozyme becomes unfolded, it can form up to three different aggregate types with characteristic peak positions/areas.

Another point worth noting is the self-replication properties of the three different structures. In the case of aggregates formed under conditions where lysozyme was predominantly folded, they were able to easily replicate their structure at the highest tested temperature (Figure 5A,D). They also formed long fibrils when replicated under quiescent conditions (Figure 4D). This suggests that they could incorporate lysozyme monomers, regardless of their folding state. Conversely, this was not the case for the aggregate types that were formed at higher temperatures. While they appeared to be able to self-replicate under their initial preparation conditions (Figure 4B,C), they were unable to do so under a temperature, where lysozyme was folded. One possibility is that such aggregate types are only capable of incorporating partially or fully unfolded protein molecules. Another likely explanation is that the higher temperature samples contained a significant number of non-amyloid structures, as seen in Type 2 and especially Type 3 AFM images (Figure 4E,F).

The final significant observation from this study is the remarkably high correlation between the structure of lysozyme aggregates and their ThT-binding characteristics. The transition from Type 1 (at 50 °C) to Type 2 and 3 (at 65 °C) structures caused a 10-fold decrease in the average fluorescence intensity values (Figure 2F). This was also followed by a shift in the bound-ThT EEM maxima positions, which signifies a different dye binding mode (Figure 2A–D) [40]. Such a massive decrease in signal value may be related to the reduction in the aggregate beta-sheet content, which was evident based on the sample FTIR spectra (Figure 3A–D) or the formation of non-amyloid structures (Figure 4E,F). The intermediate temperature aggregates also had a very large ThT fluorescence intensity value deviation, with a several-fold difference between samples from identical conditions. Taking into consideration that ThT fluorescence intensity is often used as a parameter of the relative abundance of amyloid aggregates, such variations can have drastic effects on the conclusions drawn from experimental data.

Overall, these results highlight the importance of the folding state of lysozyme on its amyloid aggregation kinetic and structural parameters, as well as their variability. Taking into account that a similar relationship exists for the neurodegenerative-disease-related prion protein, it suggests that this is a significant influencing factor for amyloid fibril formation and should be taken into account during protein aggregation studies.

## 4. Materials and Methods

### 4.1. Lysozyme Aggregation Kinetics

Hen egg-white lysozyme powder (Sigma-Aldrich, St. Louis, MO, USA, cat. No. L6876) was dissolved in 1 × PBS buffer (pH 7.4) containing 2 M guanidine hydrochloride (GuHCl) to a protein concentration of 300 µM (ε_280_ = 37,970 M^−1^cm^−1^). Thioflavin-T (ThT) powder (Sigma-Aldrich, St. Louis, MO, USA, cat. No. T3516) was dissolved in MilliQ H_2_O to a concentration of ~11 mM and filtered through a 0.22 µm pore size syringe filter. The concentration of the ThT solution was determined by diluting an aliquot of the dye solution 100 times with H_2_O and scanning its absorbance at 412 nm (ε_412_ = 23,250 M^−1^cm^−1^). The final ThT stock solution concentration was then set to 10 mM. The lysozyme, ThT and reaction buffer (1 × PBS (pH 7.4) with 2 M GuHCl) were combined to result in a solution with 200 µM lysozyme and 100 µM ThT. In order to avoid batch-to-batch variability, 80 mL of this solution was divided into 20 mL portions and frozen at −20 °C prior to use in kinetic experiments.

To measure lysozyme aggregation kinetics, 20 mL of frozen reaction solution was thawed at room temperature and distributed to a 96-well plate (200 µL final volume, each well contained one 3 mm glass bead), after which it was sealed with Nunc-sealing tape. Aggregation kinetics were monitored in a ClarioStar Plus (BMG Labtech, Ortenberg, Germany) plate reader by measuring sample fluorescence emission intensity (excitation wavelength—440 nm, emission—480 nm) every 5 min, with constant 600 RPM orbital shaking between measurements. The experiment was conducted under four different temperatures based on the lysozyme melt assay (from 50 °C to 65 °C). Reaction lag times, rates and end-point fluorescence values were determined as shown previously [27]. All data processing was done using Origin (OriginLab Corporation, Northampton, MA, USA) software.

### 4.2. Lysozyme Melt Assay

8-anilinonaphthalene-1-sulfonic acid (ANS) powder (Sigma-Aldrich, St. Louis, MO, USA, cat. No. A1028) was dissolved in MilliQ H_2_O at room temperature under dark conditions. The solution was then filtered through a 0.22 µm pore size syringe filter. The concentration of the ANS solution was determined by diluting an aliquot of the dye solution 100 times with H_2_O and scanning its absorbance at 351 nm (ε_351_ = 5100 M^−1^cm^−1^). The final ANS stock solution concentration was then set to 10 mM and stored at 4 °C under dark conditions prior to use. For the melt assay, lysozyme was prepared as described in the aggregation kinetics method section with ANS added instead of ThT. The lysozyme samples were then placed in 3 mm pathlength cuvettes (200 µL volume each) and sealed with plug caps.

The melt assay was conducted by placing the cuvettes in a Varian Cary Eclipse (Agilent, Santa Clara, CA, USA) spectrofluorometer and measuring sample fluorescence intensity (excitation wavelength—370 nm, emission—470 nm) under a range of temperatures (starting temperature—25 °C, end temperature—90 °C, incubation at starting temperature—5 min, temperature change rate—2 °C/min, measurements every 30 s).

### 4.3. Excitation–Emission Matrices

After aggregation reaction kinetic measurements, the 96-well plates were cooled down to 25 °C. In order to account for ThT hydroxylation under neutral pH and elevated temperatures, each well was supplemented with 1 µL of the ThT stock solution (additional 50 µM ThT in each sample). The plates were then placed in the plate reader and incubated at 25 °C for 10 min under constant 600 RPM shaking. Immediately after agitation, each well’s ThT excitation–emission matrix (EEM) was scanned. This was done by measuring sample fluorescence at a constant 485 nm emission wavelength under a range of excitation wavelengths (from 430 nm to 460 nm) and measuring sample fluorescence under a range of emission wavelengths (from 470 nm to 500 nm) at a constant 445 nm excitation wavelength. The data were then combined into an EEM using the ClarioStar MARS 3D spectra function. Each EEM maximum intensity position was determined by calculating the “center of mass” of the highest 10% intensity value positions as described previously [27].

### 4.4. Fourier-Transform Infrared Spectroscopy

An aliquot (100 µL) from each aggregation reaction kinetics plate well was removed and combined with 900 µL of initial reaction solution (prepared as described in the aggregation kinetics method section) and placed in a 1.5 mL test tube. Each test tube contained two 3 mm glass beads and was sealed with parafilm. The test tubes were then incubated in a dry-bath incubator under constant 600 RPM agitation at their respective temperature (based on initial sample preparation temperature). After 24 h of incubation, the samples were cooled down to room temperature and centrifuged at 12 500 RPM for 15 min. After this, the supernatant was removed and the aggregate pellets were resuspended into 500 µL of D_2_O, containing 400 mM NaCl (replacement of H_2_O with D_2_O helps to avoid H-O-H bands, which overlap with the Amide I region and the addition of NaCl improves fibril sedimentation [39]). The samples were then mixed, centrifuged, and resuspended into 200 µL D_2_O with 400 mM NaCl. The centrifugation and resuspension procedure were repeated 4 times in total. After the final supernatant removal, the fibril pellets were resuspended into 50 µL of D_2_O with 400 mM NaCl.

Samples were placed between two CaF_2_ transmission windows separated by a 0.05 mm teflon spacer. FTIR spectra were scanned using a Bruker Invenio S FTIR spectrometer, equipped with a liquid-nitrogen-cooled mercury cadmium telluride detector, at room temperature and constant dry-air purging. For every sample, 256 interferograms were recorded at 2 cm^−1^ resolution and averaged. D_2_O and water vapour spectra were subtracted from the sample spectra, which were then baseline corrected and normalised to the same band area in a range between 1700 cm^−1^ and 1590 cm^−1^. Spectra decomposition was done by using a Peak Fitting function (mixed Gaussian-Lorentzian) in a range between 1590 cm^−1^ and 1700 cm^−1^. For each spectrum, the maximum peak number was set to 6 and RMS Noise was set to 0.04 when applying the fit. Peak parameter (maximum position and area) correlation graphs were then plotted for each sample and superimposed for comparison. All data processing was done using GRAMS software.

### 4.5. Aggregate Reseeding

Based on the results of the FTIR measurements, three distinct secondary structure samples were chosen for reseeding under different temperatures. Aliquots of lysozyme aggregate samples (80 µL) were combined with the initial reaction solution (720 µL) and incubated as described in the lysozyme aggregation kinetics section (200 µL final volume with a 3 mm glass bead in each well). Seeds prepared at 50 °C were incubated at 65 °C and samples prepared at 65 °C were incubated at 50 °C. After aggregation had occurred, the samples were cooled down to 25 °C and their FTIR spectra were measured as described previously. For sample reseeding under their respective initial preparation temperatures under quiescent conditions, the procedure was done without agitation or the addition of glass beads.

### 4.6. Atomic Force Microscopy

Aliquots of samples from the reseeding experiment (30 µL, after 800 min of quiescent incubation at their respective temperatures) were gently mixed by repetitive pipetting, placed on freshly cleaved mica and left to adsorb for 3 min. The samples were then gently washed with 3 mL of H_2_O and dried using airflow. Further, 1024 × 1024 pixel three-dimensional images were obtained using a Dimension Icon atomic force microscope (Bruker, Billerica, MA, USA) as described previously [40]. Aggregate cross-sectional heights were determined by tracing line profiles perpendicular to the fibril axes. All data processing was done using Gwyddion software.

## Figures and Tables

**Figure 1 ijms-23-05421-f001:**
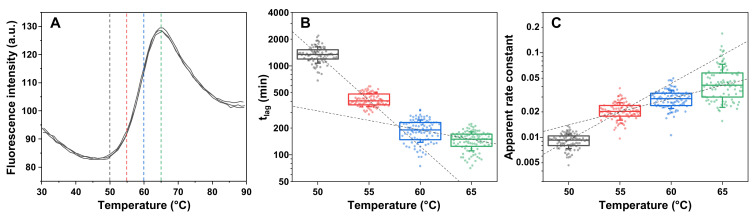
Lysozyme aggregation kinetic parameters depended on folding state. 8-anilinonaphthalene-1-sulfonic acid (ANS) signal intensity under a range of temperatures and in the presence of 200 µM lysozyme in PBS with 2 M guanidine hydrochloride (GuHCl) (pH 7.4). (**A**) Colour-coded dashed lines indicate temperatures chosen for further analysis. Lysozyme aggregation reaction t_lag_ (**B**) and apparent rate constant (**C**) values under the four selected temperature conditions (all data are colour-coded). Panel B and C box plots indicate the interquartile range and error bars are one standard deviation (*n* = 96). ANS protein melt and aggregation assay procedures are described in the Materials and Methods section. Raw data are available as Appendix A.

**Figure 2 ijms-23-05421-f002:**
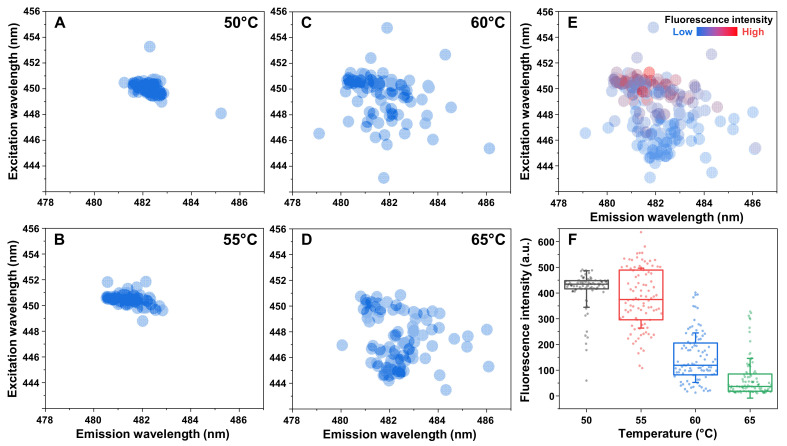
Correlation between sample fluorescence parameters and lysozyme aggregation temperatures. Bound ThT excitation-emission matrix (EEM) “center of mass“ positions of samples prepared under 50 °C (**A**), 55 °C (**B**), 60 °C (**C**) and 65 °C (**D**) conditions. EEM position and signal intensity correlation of 50 °C and 65 °C samples (**E**). Fluorescence intensity value distribution of samples prepared under different aggregation conditions (**F**). Box plots indicate the interquartile range and error bars are for one standard deviation. All data acquisition and analysis procedures are described in the Materials and Methods section. Raw data are available in the Appendix A.

**Figure 3 ijms-23-05421-f003:**
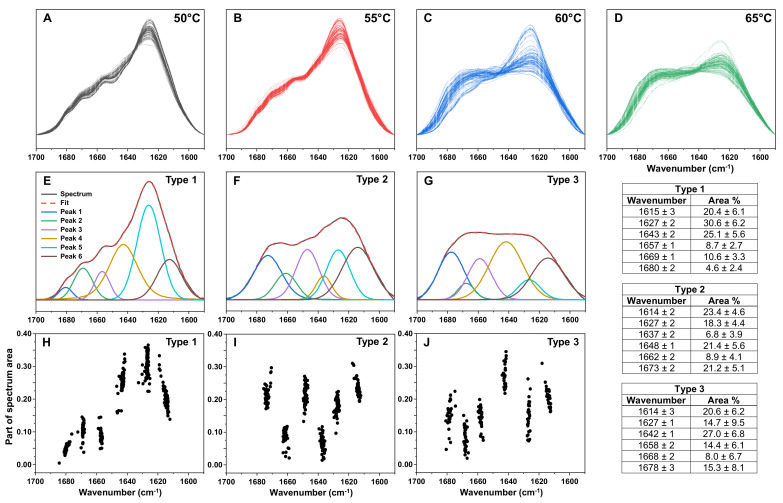
Fourier-transform infrared (FTIR) spectra of lysozyme aggregates prepared under different temperature conditions. Spectra of lysozyme aggregates prepared at 50 °C (**A**), 55 °C (**B**), 60 °C (**C**) and 65 °C (**D**), with all sample (*n* = 96) spectra superimposed and colour-coded. Peak-fit of Type 1 (**E**), Type 2 (**F**) and Type 3 (**G**) aggregate spectra and peak position/area distributions (**H**–**J**). The peak-fitting procedure was performed on 50 °C and 65 °C condition spectra. Table inserts display the peak positions and areas of all three aggregate-type FTIR spectra. Raw data are available in the Appendix A.

**Figure 4 ijms-23-05421-f004:**
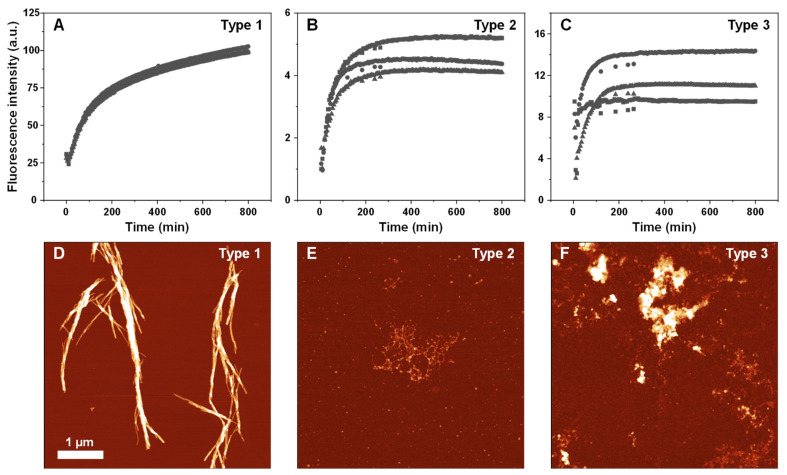
Self-replication kinetics (**A**–**C**) and atomic force microscopy images (**D**–**F**) of three types of lysozyme aggregates. Kinetic curves are from three repeats (different symbols represent separate samples). Self-replication and AFM imaging procedures are described in the Materials and Methods section. Raw data are available in the Appendix A.

**Figure 5 ijms-23-05421-f005:**
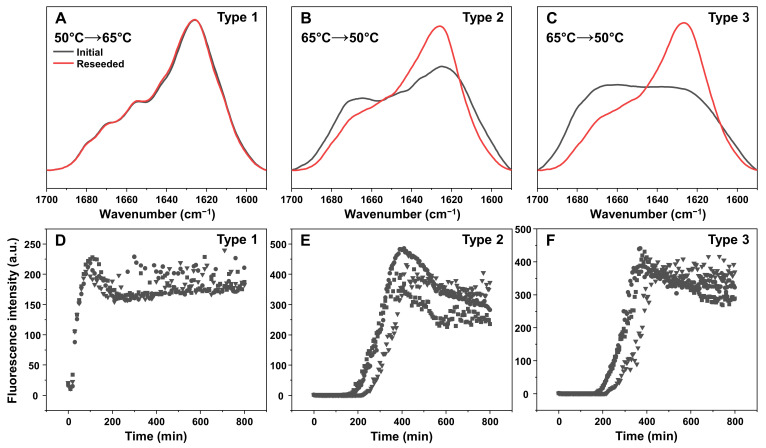
Fourier-transform infrared (FTIR) spectra of Type 1 (**A**), Type 2 (**B**) and Type 3 (**C**) lysozyme aggregates before and after the reseeding reaction and their respective aggregation kinetic curves (**D**–**F**). The reseeding procedure is described in the Materials and Methods section. Kinetic curves are from three repeats (different symbols represent separate samples). Raw data are available in the Appendix A.

## Data Availability

All data are available in the Appendix A.

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
