# Peer review of "Lysozyme Amyloid Fibril Structural Variability Dependence on Initial Protein Folding State"

_ijms, 2022, doi:10.3390/ijms23105421_

Round 1
Reviewer 1 Report
The paper "Lysozyme Amyloid Fibril Structural Variability Dependence on Initial Protein Folding State" by Mikalauskaite et al. reports an extensive biophysical investigation of the structural properties of a model protein, the lysozyme, under the conditions that promote the formation of amyloid fibrils. The authors have used in a detailed way experiments of ThT fluorescence, FTIR spectroscopy and AFM and samples have been investigated at different temperatures. Results show that the initial folding state of lysozyme, which depends on temperature, is a fundamental factor that regulates the amyloid aggregation features. At lower temperature, the cross-beta content of the aggregated species is much higher than the one at high temperature. In particular, by a detailed analysis of FTIR spectra, three patterns have been identified, each of them characterized by a different secondary structure. The results of AFM and self-replications experiments have also confirmed the importance of the initial folding state of lysozyme on its aggregation kinetic and structural properties.
Despite a huge number of papers have been written in the field of the structural investigation of amyloid fibrils, the innovative methodological approach followed by the authors is important and can be extended also to other proteins directly related to severe neurodegenerative diseases. The presentation and the discussion of the results is comprehensive and is complemented by a good quality of figures. The English is good. Overall, this work is of good quality and deserves to be published in the International Journal of Molecular Science.
However, in the opinion of this reviewer, some modifications are necessary in order to improve the readability of the paper. First, the introduction of the paper should be extended by including the contributions that other experimental techniques, such as NMR and SAXS, have provided to understand the formation and the structure of amyloid fibrils. Secondly, at the beginning of the Results section it should be stated that all details regarding sample preparation, experiments and data analysis are reported at the end of the article. Finally, the discussion of the FTIR analysis (lines 165-176) may be changed in order to give more emphasis to the determination of the amount of cross beta structures in the three types of patterns.
Minor points
Line 55: it is not clear what is “this parameter”.
Line 68: specify that PBS is a buffer.
Line 69: ANS has not been defined.
Caption of Figure 1: specify that in panels B and C the box-plots have been determined.
Line 351: a plural is missing
Figure 4, panel D. The bar length cannot be 1 m.
Line 250: the grammar of this sentence is clearly wrong.
Reviewer 2 Report
The authors investigated lysozyme fibril polymorphism when varying the temperature and initial folding state. Polymorphism is not a new topic but with growing interest. The manuscript is well written and provides a data set which might be useful for further investigation of the underlying polymorphism principles. I recommend the manuscript to publish in IJMS with my minor comments below.
What is the role of 2M GdmCl in the solution and what is the impact of the denaturant to polymorphism? Since the folded state is important to form the amyloid fibril structure of lysozyme, if the authors increase the GdmCl concentration, will lysozyme prefer to form amorphous aggregates the same as increasing the temperature?
The analysis of the FTIR seems to be important in interpreting the results. How did the authors decide how many types to include? How bad will a two-type model fit the data? From looking at the AFM image, type 2 and 3 can both be amorphous aggregates.
Have the authors tested the concentration dependent aggregation kinetics for different aggregation structures/temperatures? This might be beyond the scope of the current manuscript. It would be interesting to see whether some other mechanisms (by fitting with for instance AmyloFit) might play a role.
It's a little bit confusing at the end to extend the scope of this study to the prion like protein since most of them are intrinsically disordered without folding and unfolding. However many of them are also involved in liquid-liquid phase separation, which could play a similar role as folding and unfolding on polymorphism.
Reviewer 3 Report
This paper „Lysozyme amyloid fibril structural variability dependence on initial protein folding state“ is very interesting work, well understood and explained.
I have only minor comments:
- On page 5 is written that 1648 cm-1 is an unstructured region. Is not this the helix area? Is not this the residual α-helix of lysozyme that has not turned into beta-sheets?
- Was made AFM images of aggregates from the initial measurements? I mean from those that were not reseeded or self-replicated. Can you put them in a supplement?
- AFM images presented in the publication relate to aggregates prepared at what time? Which specific point from the kinetic curve belongs to the AFM images?
